# Additional Evidence on Serological Correlates of Protection against Measles: An Observational Cohort Study among Once Vaccinated Children Exposed to Measles

**DOI:** 10.3390/vaccines7040158

**Published:** 2019-10-22

**Authors:** Tom Woudenberg, Rob van Binnendijk, Irene Veldhuijzen, Frits Woonink, Helma Ruijs, Fiona van der Klis, Jeroen Kerkhof, Hester de Melker, Rik de Swart, Susan Hahné

**Affiliations:** 1Centre for Infectious Disease Control, Netherlands Institute for Public Health and the Environment (RIVM), Antonie van Leeuwenhoek 9, 3720 MA Bilthoven, The Netherlands; rob.van.binnendijk@rivm.nl (R.v.B.); irene.veldhuijzen@rivm.nl (I.V.); helma.ruijs@rivm.nl (H.R.); Fiona.van.der.Klis@rivm.nl (F.v.d.K.); jeroen.kerkhof@rivm.nl (J.K.); hester.de.melker@rivm.nl (H.d.M.); susan.hahne@rivm.nl (S.H.); 2Public Health Service, Region Utrecht, De Dreef 5, 3706 BR Zeist, The Netherlands; FWoonink@ggdru.nl; 3Department of Viroscience, Erasmus MC, Dr. Molewaterplein 40, 3015 GD Rotterdam, The Netherlands; r.deswart@erasmusmc.nl

**Keywords:** measles, correlate of protection, vaccination, MMR, humoral immunity

## Abstract

To assess correlates of protection against measles and against subclinical measles virus (MV) infection, we recruited once-vaccinated children from geographic regions associated with increased MV circulation and/or at schools with low vaccination coverage in the Netherlands. Paired blood samples were collected shortly after onset of the measles outbreak and after the outbreak. A questionnaire was used to document the likelihood of exposure to MV and occurrence of measles-like symptoms. All blood samples were tested for MV-specific antibodies with five different assays. Correlates of protection were assessed by considering the lowest neutralizing antibody levels in children without MV infection, and by ROC analyses. Among 91 participants, two seronegative children (2%) developed measles, and an additional 19 (23%) experienced subclinical MV infection. The correlate of protection against measles was lower than 0.345 IU/mL. We observed a decreasing attack rate of subclinical MV infection with increasing levels of specific antibodies until 2.1 IU/mL, above which no subclinical MV infections were detected. The ROC analyses found a correlate of protection of 1.71 IU/mL (95% CI 1.01–2.11) for subclinical MV infection. Our correlates of protection were consistent with previous estimates. This information supports the analyses of serosurveys to detect immunity gaps that require targeted intervention strategies.

## 1. Introduction

Measles is a highly contagious viral disease. Around 10 days after exposure, the first clinical symptoms occur, consisting of fever, cough, coryza, and conjunctivitis. About three days after the onset of fever, a maculopapular rash spreads from the face and the neck to the extremities [1]. Complications due to opportunistic pathogens most often present in the respiratory tract. Measles virus (MV) infects cells of the immune system and causes lymphopenia leading to immunosuppression, which can last up to two to three years, leaving individuals vulnerable to secondary infections [2,3].

Measles vaccination programs have led to a significant decrease in measles incidence, resulting in corresponding reductions in measles mortality and morbidity [4]. Worldwide coverage with the first dose of a measles-containing vaccine increased to around 85% but stabilized since 2009. As the incidence of measles is decreasing, population immunity will gradually become more dependent on vaccine-induced immunity. 

Vaccine-induced immunity provides lower measles antibody concentrations than naturally induced immunity [5,6,7,8]. While natural infection has been shown to provide life-long immunity [9], 2–10% of individuals with vaccine-induced immunity may not develop or sustain protective humoral immunity [10,11,12]. As a result, outbreaks of measles, generally initiated by unvaccinated index cases, have been observed in vaccinated populations [13,14,15]. In vaccinated subjects, clinical symptoms following MV infection may range from being absent (subclinical MV infection associated with secondary immune responses) to full-blown measles. MV transmission from twice-vaccinated individuals has rarely been observed [13,16]. From the perspective of measles elimination, monitoring the immunity against measles of vaccinated populations is essential.

Immunity against measles consists of both humoral and cellular immune responses. Humoral immunity is mostly involved in the prevention of MV infection, whilst cell-mediated immunity is required to clear the virus once infection has occurred. Tests for humoral immunity are more widely available and standardized than those for cellular immunity and are therefore most often used to assess measles immunity [17]. The presence of neutralizing antibodies, commonly demonstrated by the plaque reduction neutralization, is considered the most reliable assay for serological immunity [18]. 

Measuring antibody levels prior to and after exposure to MV combined with surveillance of measles has provided some insight in the correlate of protection, i.e. the antibody level needed to prevent against disease or infection [19]. However, this evidence is limited to a few studies with a small number of participants. In the US [20], seven out of nine students with neutralizing antibody concentrations equivalent to values below 0.12 IU/mL measured prior to an outbreak developed measles compared with none of 71 students with titers above 0.12 IU/mL. In Senegal, where a measles outbreak occurred during a vaccine trial, 13 of 36 (36%) children with titers of 0.04 to 0.125 IU/mL developed measles, compared with seven out of 258 (3%) of those with titers higher than 0.125 IU/mL [21]. However, the authors noted that many seronegative vaccinated children were also protected against measles, most likely indicating the presence of cellular immunity [21]. In the Netherlands, during a measles outbreak among health care personnel, antibody titers up to 0.146 IU/mL were insufficient to protect against measles [14]. 

Correlates of protection against subclinical infection (as measured by secondary immune responses) were studied during an outbreak among children in Luxembourg. Exposed parents who previously experienced measles were boosted when their pre-exposure neutralization titers were below 64. None of these parents reported any symptoms [22]. Among vaccinated individuals, estimates of correlates of protection against subclinical MV infection range from approximately 1.0 IU/mL [20,23] to 4.0 IU/mL [21].

In the Netherlands, immunization programs against measles were introduced in 1976 (monovalent measles vaccine), followed by the introduction of measles-mumps-rubella (MMR) vaccination in 1987. The first MMR is given to infants at 14 months of age, and the second dose to children at nine years of age. Vaccination coverage was around 95% for a significant period of time for the first dose [24], and measles has become a rare disease. However, large outbreaks continue to occur among unvaccinated children in the Orthodox Protestant community, as observed in 1983, 1988, 1993, 1999/2000, and 2013/2014 [25,26]. Orthodox Protestants form a socially and geographically clustered minority group in the Netherlands of about 250,000 individuals, among whom vaccination coverage is approximately 60% [27]. Foreseeing an outbreak based on a serosurvey and mathematical modelling [5,28], we designed a study to assess immunological correlates of protection against measles, assuming that vaccinated children attending schools with low vaccination coverage would likely be exposed to MV. The outbreak, which included an estimated 30,000 measles cases, started in May 2013 and lasted until March 2014 [29]. The main objective of our study was to identify serological correlates of protection against measles and MV infection among once vaccinated children. 

## 2. Materials and Methods

### 2.1. Study Design

We performed an observational cohort study among once vaccinated children aged 4–8 years during the 2013–2014 measles epidemic in the Netherlands. Pairwise blood samples were collected. Four serological tests were used to determine MV infection. Neutralizing antibody concentrations were determined in the fifth assay to assess the correlate of protection.

### 2.2. Participants

Eligible children had received one MMR vaccination, M-M-RVAXPRO (Merck & Co., Inc. Kenilworth, NJ, USA) containing more attenuated MV Enders’ Edmonston strain and were enrolled in an Orthodox Protestant primary school. Children who received a second MMR during the study period were excluded from the analyses. Eligible children were invited by two approaches. The first approach consisted of the identification of MMR-1 vaccinated children of four to eight years of age from the national vaccination register resident in municipalities with Orthodox Protestant schools. These municipalities were chosen to ensure that children had a probability to be enrolled in an Orthodox Protestant primary school. After the number of participants from the first approach remained unsatisfactory low, an additional approach was used, in which participants were invited directly via Orthodox Protestant primary schools. This was only feasible when the management team of the school gave permission for this to the local municipal health service. 

### 2.3. Data Collection

Parents of all potentially eligible children were sent an invitation letter, an informed consent form, and a questionnaire. The latter ascertained their eligibility and consisted of questions including sex, date of birth, and whether the child experienced measles in the past. Parents of eligible children were subsequently invited to attend a clinic where their child was asked to give a blood sample through a venipuncture. This blood sample was taken shortly after onset of the outbreak.

After March 2014 when the outbreak had ended [29], children were invited to attend the clinic for a finger stick blood sample. Sera were heat-inactivated for 30 min and subsequently stored at −20 °C until use. Parents of children were then asked to fill in a second questionnaire, which documented potential exposure to MV and occurrence of clinical symptoms potentially related to MV infection (rash, fever, cough, conjunctivitis, sore throat, coryza, Koplik spots, headache, listlessness, vomiting, diarrhea, swollen glands in neck) in the period between the two blood samplings. We defined exposure to MV at school by measles cases reported from the school to the national register of notifiable diseases during the measles outbreak. Exposure to MV elsewhere was ascertained from information provided by parents in the questionnaire. Children who were enrolled in an Orthodox Protestant school without reported measles cases and who were not exposed to measles by parental recall were excluded from all analyses.

### 2.4. Laboratory Tests

All serum samples were tested pairwise for measles specific antibodies with five serological assays—a bead-based multiplex immunoassay (MIA) for total MV specific IgG [30], an immunofluorescence assay to detect antibody levels specific for MV-F (FIgG) or MV-H (HIgG) [31], an indirect EIA to detect antibodies to MV-N (NIgG) [32], and by an in-house focus reduction neutralization test (FRNT). Laboratory tests are further specified in Appendix A.

### 2.5. Case Classification

We decided a priori that children could be classified into the following three classes: those having had MV infection prior to the study period, MV infection during the study period, or no MV infection. We defined clinical measles by fever, rash, and at least one out of cough, coryza, and conjunctivitis, as reported in the questionnaire filled in by parents [33]. We identified MV infections by calculating the ratio between pre- and post-test results of four out of the five immunoassays used (the MIA, HIgG, FIgG, and NIgG). The fifth assay (FRNT) was used independently to assess the correlates of protection. The 10log-transformed normalized ratios of the four immunoassays were used to classify children using k-means cluster analyses. This involves applying an algorithm to differentiate groups while minimizing the within-cluster sum of squares [34]. Normalizing the ratios ensured that the deviation from the average per sample was considered equally for all four immunoassays in the k-means cluster analyses. 

### 2.6. Statistical Methods

We estimated the overall attack rate of MV infection by including infections occurring prior to and during the study period. Univariable logistic regression was used to assess determinants of MV infection and to determine whether the occurrence of symptoms differed between those who experienced MV infection during the study period and those who did not. 

Correlates of protection against measles or subclinical MV infection were assessed among all participants except those who experienced MV infection prior to the first sampling. First, we considered as the correlate of protection the lowest concentration in pre-sera among children who did not develop clinical measles or subclinical MV infection during study period. Second, we assessed the correlates of protection using receiver operator characteristics (ROC) analyses. The FRNT antibody level corresponding with the highest sum of the sensitivity and specificity on the ROC curve was considered the correlate of protection. The area under the curve (AUC) was estimated as previously reported [35]. We also assessed whether a relationship existed between pre-exposure antibody concentrations and the attack rate of MV infection during follow up using Fisher’s exact test.

An assumption in the assessment of the correlate of protection is that all children were exposed during the outbreak. We performed a sensitivity analysis to assess whether our estimates would hold with a selection of children who were most likely exposed (enrolled in a school with reported measles cases and exposed to measles according to their parents).

Data visualization and data analyses were carried out using R (Version 3.4.0, R Foundation for Statistical Computing, Vienna, Austria). Package “pROC” was used to visualize the ROC curve and to assess the optimal cut-off [36]. 

### 2.7. Ethics Statement

The Central Committee on Research involving Human Subjects (CCMO) provided ethical permission to perform the study (CCMO 13.0520). Informed consent was obtained from the parent(s) of the children. 

## 3. Results

Of 13,344 parents invited through the national vaccination register, 2579 submitted the initial questionnaire. Of these, 279 of their children were eligible and were invited for the first blood sampling. Blood samples were taken from 27 children. Via Orthodox Protestant schools, parents of 738 children were invited to participate. Blood samples were collected from 92 children. In total, 119 children were enrolled in the study. Of these, 28 children were excluded from the analyses—16 because they received a second MMR during study period, three because they did not participate to the second sampling, and nine because they were not enrolled in an Orthodox Protestant school with reported cases of measles nor experienced exposure to MV according to their parents. Thus, the analyses included 91 children, of whom 21 were enrolled via the national vaccination register and 70 via Orthodox Protestant schools. 

### 3.1. Descriptive Results

Of the 91 participants, 41 were boys (Table 1). Median age at the first sampling was 6.5 years (IQR 5.5–7.5). Median follow-up period was 8.4 months (IQR 6.6–8.4). The distribution of antibody concentrations measured with the four immunoassays of the first blood sampling against the ratio between the first and second sampling are shown in Figure 1. From the 10log-transformed normalized ratios deduced from pre- and post-scores of four immunoassays, we instructed the k-means clustering algorithm to identify three groups. One group consisted of eight children with relatively low ratios indicating MV infection prior to study period. Another group consisted of thirteen children with relatively high ratios indicating MV infection during study period. Children with ratios around the value of 1 (*n* = 70) were assigned to the group no MV infection. The overall attack rate of MV infection in the study sample was 23% (21/91). The attack rate was 22% (18/92) among children with high exposure to measles (enrolled in a school with reported cases and exposure according to the parents) and 33% (3/9) with medium exposure to measles (enrolled in a school without reported measles cases but with exposure according to the parents). Sex, age, and moment of inclusion were not predictive of the attack rate. 

Two children developed symptomatic measles during the study period. Both were also retrieved as reported cases in the Dutch national register of notifiable infectious diseases. These two children had no detectable virus neutralizing antibodies at the first sampling (≤0.06 IU/mL) and had neutralizing antibody levels of 2.96 IU/mL and 6.40 IU/mL at the second sampling. 

Those who experienced subclinical MV infection during the study period (*n* = 11) had antibody concentrations ranging from 0.345 IU/mL to 2.060 IU/mL in the FRNT assay in their first sample. The following symptoms were reported among these 11 children during study period: rash (0 children), fever (three children), cough (two children), conjunctivitis (one child), and coryza (two children). These children did not differ with regard to the frequency of reported measles compatible symptoms compared with children who did not experience MV infection. 

### 3.2. Correlates of Protection

Three children had no detectable (neutralizing) antibodies in their first blood sample (Figure 2). Two of these developed measles including seroconversion. No measles was observed in participants other than these two. We consider these children to have had primary vaccine failure of the first measles vaccination. Due to the low number of measles cases, we unfortunately could not assess the correlate of protection using a ROC curve nor the relationship between the attack rates and neutralizing antibody levels. The lowest measurable FRNT concentration in pre-sera of children without measles during study period was 0.345 IU/mL (dashed line in Figure 2). 

The lowest concentration of FRNT antibodies observed above which no MV infection was observed among children was 2.06 IU/mL (dotted line in Figure 2). The ROC analyses indicated that the sum of the sensitivity and specificity was highest at a correlate of protection of 1.71 IU/mL (95%CI 1.01–2.11 IU/mL) against MV infection, which corresponds to a sensitivity of 92% (95% CI 77–100) and a specificity of 59% (95% CI 43–78) in our study population (Figure 3). The lower value derived from the ROC analyses results from the optimization for both the sensitivity and the specificity whereas the other approach seeks a sensitivity of 100%. The AUC (area under the curve) was 0.76 (95% CI: 0.65–0.88). The attack rate for MV infection was inversely related to the antibody concentrations measured before exposure (*p* < 0.001, Fisher’s Exact test) (Table 2), although the attack rate was approximately similar between children with antibody concentrations ranging from 0.345–1.205 IU/mL and 1.206–2.540 IU/mL.

### 3.3. Sensitivity Analysis

Limiting the analyses to children who were enrolled in a school with reported measles cases and experienced exposure to MV according to parental recall (*n* = 75), we found that the correlate of protection against measles (lower than 0.345 IU/mL) or MV infection (2.1 IU/mL) remained the same. The correlate of protection against MV infection found with the ROC decreased slightly to 1.59 IU/mL (95% CI 0.58–2.14 IU/mL), which corresponded with a sensitivity of 91% (95% CI 64–100) and specificity of 61% (95% CI 44–94). 

## 4. Discussion

An anticipated large measles outbreak in the Netherlands provided a unique opportunity to assess correlates of protection against measles and subclinical MV infection, an area for which existing evidence is scant. 

Two out of three children who tested negative developed measles. None of the children with detectable antibodies developed measles. All of these had antibody levels above the previously established correlate of protection of 0.12 IU/mL [20]. The lowest antibody concentration observed among children with detectable antibodies was 0.345 IU/mL. Lower concentrations may still provide clinical protection, but children with these concentrations were unfortunately not present in our study. 

We found that antibody concentrations of 2.1 IU/mL and above completely protected against MV infection. This was substantially higher than the neutralizing titers of approximately 1.0 IU/mL found to prevent MV infections among students in the US and Taiwan [20,23]. However, antibody titers of approximately 4.0 IU/mL (HI titers >1:256) were needed to protect children in Senegal against MV infection [21]. These differences can be caused by differences in the intensity of exposure to MV [37,38]. Differences in neutralization assays that have been used in the past further complicate the comparison of correlates of protection [39]. The recent standardization of the neutralization assay greatly facilitated the comparison between different studies [18], but most of the previous studies that assessed correlates of protection lacked standardization [20,21]. 

The relationship observed here and in other studies [21,23,40] between antibody levels and MV infection attack rates underlines that correlates of protection against measles are relative rather than absolute. They depend on the level of exposure to measles virus and presence of cellular immunity. Cellular immunity is thought to be protective in individuals with low levels of antibodies [21,41], although it is considered to control and/or eliminate virus-infected cells rather than blocking infection [42]. T cells could therefore have played a role in preventing the occurrence of symptoms among those who experienced boosting of antibodies in our study population. 

Symptoms reported by those who experienced specific boosting of antibodies did not differ from those without antibody boosting. This suggests that the boosting of antibodies we observed was caused by a subclinical infection. 

Three of the children in our cohort had not responded to the first immunization, two of which developed measles during the outbreak, and one that will likely experience a primary immune response with the second MCV at nine years of age. For those who already showed a primary immune response, a second vaccine dose will result in only a transient increase in the antibodies [7,10,43]. 

One limitation of our study was the laborious enrollment of children, which resulted in low number of respondents and a delayed enrollment to such an extent that children were enrolled after the onset of the outbreak. As a result, some participants already experienced MV infection prior to their inclusion in our study. However, we were fortunate to identify the infected individuals well by measuring significant antibody decay. Timing between the first and second blood sampling is an important prerequisite to measure this decay, as was recently shown by others [44] and us [14], which enabled us to distinguish those who experienced regular waning immunity of vaccine-induced immunity from those that experienced a steep decay indicative of a recent infection. 

The low response rate limited the possibilities and precision of estimating correlates of protection. However, we do not think that the low response biased our results—we have no reason to believe that those that participated were different from non-participants in terms of exposure to MV or measles immunity.

Another limitation is the assumption that all children were exposed to measles. As one out of three fully susceptible children did not develop measles, this assumption was not entirely correct. Yet, the attack rate of MV infection in our study was high and our study participants were enrolled in Orthodox Protestant primary schools. As Orthodox Protestants between four and 12 years of age were the most affected group during an outbreak of about 30,000 MV infections [45], we can assume that the majority in our study population was indeed exposed to measles. Furthermore, the results did not change when we limited the analyses to children with highest likelihood of exposure to MV. 

To prevent outbreaks among high vaccination coverage populations, immunity gaps can be found by monitoring antibodies in populations to guide the implementation of immunization strategies. High vaccination coverage alone does not guarantee adequate population immunity due to for example gaps in cold-chain quality or waning immunity and should be supported by serosurveys. Our new evidence about the level of antibodies that are protective against measles and MV infection is crucial here.

## Figures and Tables

**Figure 1 vaccines-07-00158-f001:**
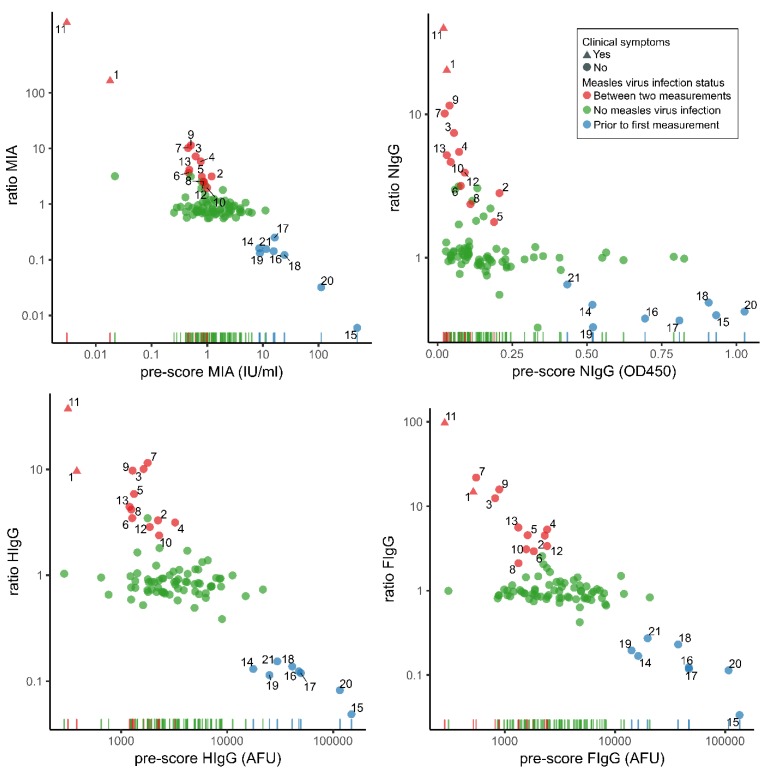
Ratios of pre- and post-measurements of measles specific antibody concentrations by pre-outbreak results of 91 children. The colors indicate the classification based on the k-means clustering analyses. Numbers provide a comparison of samples across the different tests. MIA: bead-based multiplex immunoassay; FIgG: immunofluorescence assay to detect antibody levels specific for MV-F protein; AFU: arbitrary fluorescence units; HIgG: immunofluorescence assay to detect antibody levels specific for MV-H protein; NIgG: indirect EIA to detect antibodies to MV-N protein.

**Figure 2 vaccines-07-00158-f002:**
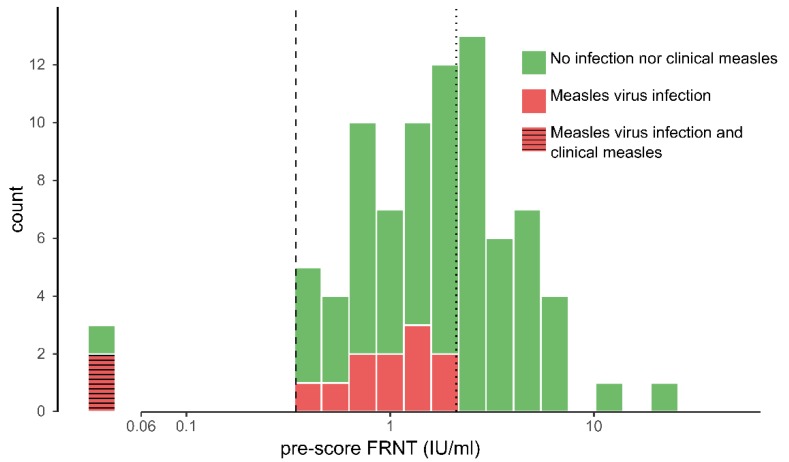
Distribution of FRNT log antibody concentrations at the first sampling in participants excluding those with evidence of measles virus (MV) infection prior to the first sample (*n* = 83) taken shortly after the onset of a measles outbreak in the Netherlands, 2013–2014. Colors indicate MV infection status. The vertical dashed line depicts the correlate of protection against measles (0.345 IU/mL) and the vertical dotted line the correlate of protection against MV infection (2.06 IU/mL). Three children had antibody concentrations below the lower limit of detection (0.06 IU/mL). FRNT: focus reduction neutralization test.

**Figure 3 vaccines-07-00158-f003:**
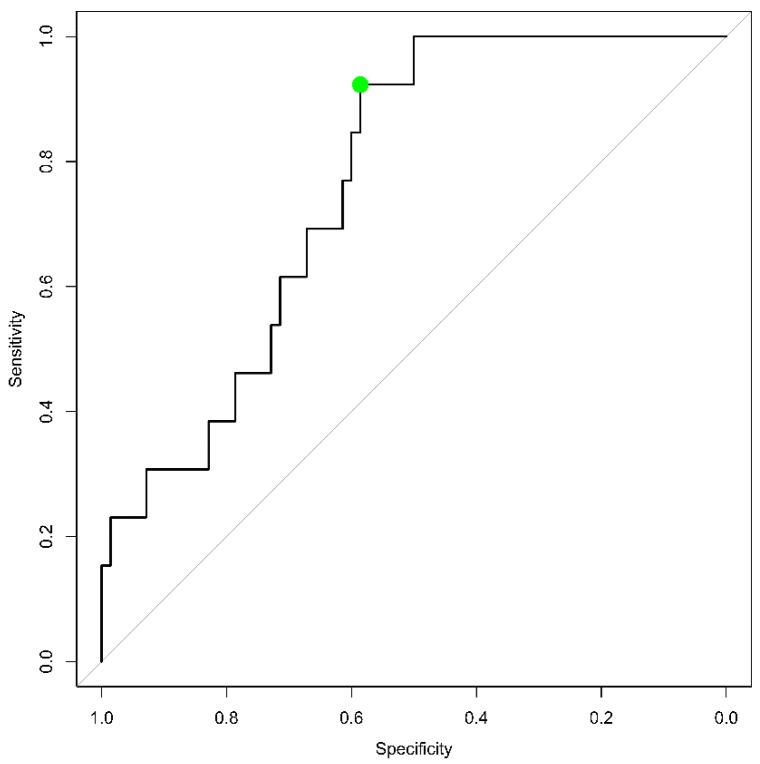
Receiver operator characteristic of the predictive value of measles neutralizing antibody concentrations measured prior to the measles outbreak in the Netherlands, 2013–2014, to protect against MV infection. The green dot corresponds with antibody levels of 1.71 IU/mL and yields the highest sum of the sensitivity and specificity. The AUC is 0.76 (95% CI: 0.65–0.88).

**Table 1 vaccines-07-00158-t001:** Characteristics of once-vaccinated participants (*n* = 91) included in an observational cohort study to assess correlates of protection against measles, the Netherlands, 2013–2014.

Characteristic	N (%)
Sex	
Boy	41 (45)
Girl	50 (55)
Enrollment	
School	70 (77)
National vaccination register	21 (23)
Age at first MMR in months (IQR)	14.5 (14.4–15.1)
Median follow-up time in months (IQR)	8.4 (6.6–8.4)
Self-reported symptoms along the follow–up	
Fever	38 (42)
Rash	3 (3)
Cough	46 (51)
Runny nose	42 (46)
Conjunctivitis	8 (9)
Exposure to MV *****	
High	82 (90)
Medium	9 (10)
Age at first sampling in years (IQR)	6.5 (5.5–7.5)
Age at second sampling in years (IQR)	7.3 (6.2–8.2)

* Exposure to MV was divided into two categories. Category “high” comprised children enrolled in a school with reported cases and exposure according to the parents. Children in category “medium” were enrolled in a school without reported measles cases but with exposure according to the parents.

**Table 2 vaccines-07-00158-t002:** Attack rates of MV infection among participants (excluding those with evidence of MV infection prior to follow-up (*n* = 8)) stratified by pre-exposure neutralization antibody concentration (*n* = 83) during a measles outbreak in the Netherlands, 2013–2014.

Antibody Concentrations Prior to Exposure (IU/mL)	Attack Rate (Cases/Exposed) (%)
≤0.06	67 (2/3)
0.35–1.21	22 (6/27)
1.21–2.54	19 (5/27)
>2.54	0 (0/26)
Total	16 (13/83)

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
