# Peer review of "Additional Evidence on Serological Correlates of Protection against Measles: An Observational Cohort Study among Once Vaccinated Children Exposed to Measles"

_vaccines, 2019, doi:10.3390/vaccines7040158_

Round 1

Reviewer 1 Report

In the present article, the authors try to determine the serological correlates of protection against measles and against “subclinical” measles virus infection in children having received 1 MMR dose.

In the actual epidemiological context of measles this is an important question. In countries with high vaccination coverage.

The administration of the second dose of MMR vaccine in Netherland, 5 years after the administration of the first one, 4 and 9 years old, furnish a window of time long enough to carry out the study. This opportunity is associated to the identification and a rigorous surveillance in regard to measles of a community of people with a very low vaccination coverage, reported here as 60%.

Comments

The subclinical measles infection is defined here by an increase of the titer of specific antibodies. In some children, other respiratory symptoms or fever, without rash, were reported.

Authors assume that these clinical situation are due to the subclinical measles virus infection. It could be a real asymptomatic measles infection or a nonspecific increase of Ab titers due to a different virus infection. It is important to discuss this aspect.

Why other virus infections, like respiratory viruses, were not investigate in children with an increase measles antibodies titer .

Also, to establish a correlate of protection, other studies including more vaccinated patients developing symptomatic measles infection should be done.

Author Response

Reviewer 1

In the present article, the authors try to determine the serological correlates of protection against measles and against “subclinical” measles virus infection in children having received 1 MMR dose.

In the actual epidemiological context of measles this is an important question. In countries with high vaccination coverage.

The administration of the second dose of MMR vaccine in Netherland, 5 years after the administration of the first one, 4 and 9 years old, furnish a window of time long enough to carry out the study. This opportunity is associated to the identification and a rigorous surveillance in regard to measles of a community of people with a very low vaccination coverage, reported here as 60%.

Comments

The subclinical measles infection is defined here by an increase of the titer of specific antibodies. In some children, other respiratory symptoms or fever, without rash, were reported.

Authors assume that these clinical situation are due to the subclinical measles virus infection. It could be a real asymptomatic measles infection or a nonspecific increase of Ab titers due to a different virus infection. It is important to discuss this aspect.

Response 1. We compared the frequency of measles compatible symptoms of children with a subclinical measles infection with children without any measles infection (i.e. children with ratios of 1 derived from the post- and pre-measurement of antibody concentrations) and we found no significant difference. This suggests that children with subclinical measles did not experience classical measles.

Why other virus infections, like respiratory viruses, were not investigate in children with an increase measles antibodies titer .

Response 2. We believe that the measurement of measles-specific antibodies is influenced minimally by immune responses targeted at other respiratory viruses and therefore irrelevant to measure in our study.

Also, to establish a correlate of protection, other studies including more vaccinated patients developing symptomatic measles infection should be done.

Response 3. We completely agree with the reviewer. Given the lack of estimates, new studies need to be conducted and if possible should aim for a bigger sample size to obtain estimates that are more robust.

Reviewer 2 Report

The authors investigated the measles PRNT levels correlated to the symptomatic measles infection, subclinical infection, and no infection. Pre-examination PRNT titer <0.345 IU/mL was related to clinically apparent measles infection, and PRNT >2.06 IU/mL completely protect against measles infection. I have several comments.

They examined the MIA, and IgG EIA against HA, F, and N proteins and the results were shown in Figure 1. Two cases No. 1 and 11 were clinically apparent measles, 11 cases were subclinical infection, and 8 were past measles infection. Ratios of pre and post titers are shown depending on the pre-titers. There was no interpretation regarding these examinations. From the results of four antibody tests, pre-examination antibody titers of MIA and IgG antibody against HA seemed to relate to clinical, subclinical, and post measles infections but not for the IgG EIA against N and F proteins. These MIA and IgG antibody against HA assays were relevant to PRNT.   After March 2014 when the outbreak was ended, the subjects were enrolled. Two cases of measles and 11 subclinical infections were reported during the study period. Measles outbreak was not ended. Surveillance data before and after March should be added. It may relate to the exposure to measles. In Table 1, 82 were high and 9 were medium exposure groups. How many were infected clinically or subclinically? In Table 2, data was similar to the content of Figure 2. The stratification of pre-exposure PRNT was shown; <0.06, 0.35-1.21, 1.21-2.53, >2.54. Where from the margin of 1.21, and 2.53?

Author Response

The authors investigated the measles PRNT levels correlated to the symptomatic measles infection, subclinical infection, and no infection. Pre-examination PRNT titer <0.345 IU/mL was related to clinically apparent measles infection, and PRNT >2.06 IU/mL completely protect against measles infection. I have several comments.

They examined the MIA, and IgG EIA against HA, F, and N proteins and the results were shown in Figure 1. Two cases No. 1 and 11 were clinically apparent measles, 11 cases were subclinical infection, and 8 were past measles infection. Ratios of pre and post titers are shown depending on the pre-titers. There was no interpretation regarding these examinations. From the results of four antibody tests, pre-examination antibody titers of MIA and IgG antibody against HA seemed to relate to clinical, subclinical, and post measles infections but not for the IgG EIA against N and F proteins (1). These MIA and IgG antibody against HA assays were relevant to PRNT.   After March 2014 when the outbreak was ended, the subjects were enrolled (2). Two cases of measles and 11 subclinical infections were reported during the study period. Measles outbreak was not ended (2). Surveillance data before and after March should be added. It may relate to the exposure to measles (2). In Table 1, 82 were high and 9 were medium exposure groups. How many were infected clinically or subclinically? (3) In Table 2, data was similar to the content of Figure 2. The stratification of pre-exposure PRNT was shown; <0.06, 0.35-1.21, 1.21-2.53, >2.54. Where from the margin of 1.21, and 2.53? (4)

We thank the reviewer for the critical evaluation of our work.

Response 1. The different tests were used to categorize children into three distinct groups depending on their measles infection status (prior, or during the follow-up or none). Figure 1 shows the results. We agree that the distinction between the groups varies among the different tests, but find that in general the distinction is quite convincing. Combining the information of four different tests that test different measles-specific antibodies reduces the inclusion of accidental outliers and thus the influence of measurement bias. Response 2. For clarification: all subjects were enrolled shortly after the onset of the outbreak. A second blood sample was taken after the outbreak. In the introduction, we report the course of the outbreak “started in May 2013 and lasted until March 2014 [29]”. Citation 29 refers to an article (by us) that describes the outbreak in detail. Adding surveillance data after the second blood samples were taken is irrelevant to the outcomes of our study.

Response 3. We find this relevant and added this information to the manuscript in lines 245-249: “The attack rate was 22% (18/92) among children with high exposure to measles (enrolled in a school with reported cases and exposure according to the parents) and 33% (3/9) with medium exposure to measles (enrolled in a school without reported measles cases but with exposure according to the parents).”.

Response 4. Except for the three individuals that were seronegative, we created groups of equal size.

Reviewer 3 Report

Overall, the manuscript by Woudenberg et al is an interesting study. The findings provide an analysis and correlation of protection against measles and subclinical measles virus infection in once–vaccinated children from a geographic region in Netherland associated with low vaccination coverage and increased virus circulation. The manuscript is well written and the results and conclusions are well presented. I only have a couple minor comments.

- Lines 84-87. The authors claim that the tests for humoral immunity are widely available and standardized, and that neutralization assay by plaque reduction is the most reliable. The author should add some reference to their statements.

- In the material and methods the authors should describe the sample collection, preparation and storage.

Author Response

Overall, the manuscript by Woudenberg et al is an interesting study. The findings provide an analysis and correlation of protection against measles and subclinical measles virus infection in once–vaccinated children from a geographic region in Netherland associated with low vaccination coverage and increased virus circulation. The manuscript is well written and the results and conclusions are well presented. I only have a couple minor comments.

- Lines 84-87. The authors claim that the tests for humoral immunity are widely available and standardized, and that neutralization assay by plaque reduction is the most reliable. The author should add some reference to their statements.

Response 1. Thank you for your diligence. We have added a reference. (REF 17: Strebel, P.M.; Papania, M.J.; Gastañaduy, P.A.; Goodson, J.L. Chapter 37 - Measles Vaccines.). In this reference one can find this statement: “Laboratory evidence of immunity is most conveniently documented by use of antibody assays because tests for cell-mediated immunity are not standardized”.

- In the material and methods the authors should describe the sample collection, preparation and storage.

Response 2. We agree with the reviewer and have described this in lines 155-156 (“Sera were heat-inactived for 30 minutes, and subsequently stored at -20 degrees until use.”).

Reviewer 4 Report

This study is very well planned, written and implemented. The results are of practical importance (detection of low levels of immunization which predict possible MV infection). The values are normalized enabling comparisons between studies and routine screening.

Author Response

We would like to thank the reviewer for her/his kind words. 

Reviewer 5 Report

Line 38: Define ROC at the first mention

Line 115: replace the period with a comma, to read “250,000”

Line 129: List the four testes. You may phrase it like, “Four serological tests, namely……..and…..were used…”

Lines 220-230: How did the authors estimate/calculate the appropriate sample size, and lies 221-224, how did the authors decide on drawing blood samples from only 27 out of 229 and 92 out of 738 children?

Author Response

Line 38: Define ROC at the first mention

Response 1. Given the limited number of words in the introduction we chose to use the abbreviation. ROC is defined in lines 200-201.

Line 115: replace the period with a comma, to read “250,000”

Response 2. Thank you, we replaced the period with a comma.

Line 129: List the four testes. You may phrase it like, “Four serological tests, namely……..and…..were used…”

Response 3. The first alinea of the methods gives an overview of the study. We prefer to go into more detail about the serological tests in section “laboratory tests” and in the supplement.

Lines 220-230: How did the authors estimate/calculate the appropriate sample size, and lies 221-224, how did the authors decide on drawing blood samples from only 27 out of 229 and 92 out of 738 children?

Response 4. Unfortunately, the reduced number of participants was due to a disappointing response rate and was not decided on by the authors.